# Early Postnatally Induced Conditional Reelin Deficiency Causes Malformations of Hippocampal Neurons

**DOI:** 10.3390/biom15121662

**Published:** 2025-11-28

**Authors:** Mária Schneider-Lódi, Ala Ahrari, Maurice Meseke, Franco Corvace, Marie-Luise Kümmel, Anne-Kathrin Trampe, Mohammad I. K. Hamad, Eckart Förster

**Affiliations:** 1Department of Neuroanatomy and Molecular Brain Research, Institute of Anatomy, Ruhr University Bochum, 44801 Bochum, Germany; maria.schneider-lodi@ruhr-uni-bochum.de (M.S.-L.); ala.ahrari@dzne.de (A.A.); maurice.meseke@rub.de (M.M.); franco.corvace@rub.de (F.C.); marie-luise.kuemmel@rub.de (M.-L.K.);; 2Nuclear Function Group, German Center of Neurodegenerative Diseases (DZNE), 53127 Bonn, Germany; 3Department of Anatomy, College of Medicine and Health Sciences, United Arab Emirates University, Al Ain 17666, United Arab Emirates; m.hamad@uaeu.ac.ae

**Keywords:** reelin, knock-out, hippocampus, granule cells, pyramidal cells, interneurons, silver staining, neuron reconstruction, dendritic morphology

## Abstract

The extracellular matrix protein reelin is well known for orchestrating radial migration of cortical neurons during embryonic cortical development. While in the *reeler* mutant mouse, lacking reelin expression, radially migrating neurons are malpositioned and display dendritic malformations, no such deficits were found after conditionally induced reelin deficiency (Reln^cKO^) in the hippocampus of mice aged two months. Here, we addressed the question whether or not Reln^cKO^, when induced early after birth, might cause malformations of hippocampal neurons. For instance, we could recently show dendritic hypertrophy of somatosensory and entorhinal cortex neurons after early induced Reln^cKO^. In the present study, reelin deficiency in Reln^cKO^ mice was induced immediately after birth, and the analysis of reconstructed Golgi-stained hippocampal neurons from these mice, when aged 4 weeks, revealed morphological malformations. Dentate granule cells were the most affected from all analyzed hippocampal neuronal cell types. Thus, Reln^cKO^ granule cells had a significantly smaller soma size and displayed atrophy of proximal dendritic segments when compared to wild type (wt). Malformations of interneurons were only subtle and cell type specific; thus, multipolar but not bitufted interneurons developed proximal dendritic hypertrophy. Also, the dendrite morphology of CA2- and CA3-pyramidal cells was affected, while we did not detect morphological changes of CA1-pyramidal cell dendrites. In summary, our results show that early postnatal Reln^cKO^ causes morphological malformations of hippocampal neurons, in particular of dentate granule cells. Taken together with our previous findings, we conclude that not only specific types of entorhinal- and neocortical neurons, but also types of hippocampal neurons are at risk of developing malformations if reelin expression is reduced during a critical early postnatal period.

## 1. Introduction

The spontaneous mutation of the reelin gene in mice was found to induce a neurological phenotype with characteristic neurological deficits that include ataxia, tremor, and a characteristic reeling gait, the latter being eponymous for the term *reeler* mutant mouse (*reeler*) [1,2]. Beyond these behavioral abnormalities, multiple region-specific neuronal migration defects, neuron-type specific malformations, and fiber tracts were found in different brain areas, including neocortex, hippocampus, cerebellum, and mesencephalon [2,3,4,5].

In *reeler*, the earliest defects in the developing neocortex are observed around embryonic day 13 (E13) during preplate splitting [6,7] and at E14, when the cortical plate (CP) emerges [3,4,8]. Initial studies revealed that dendritic processes of neurons in the CP of the *reeler* are shorter; neurons are malpositioned and not as densely packed when compared to wt animals [3,4]. Almost three decades later, reelin, expressed and secreted by Cajal–Retzius (CR) cells in the cortical marginal zone, was identified as the protein that is missing in *reeler*, its absence causing morphological and behavioral deficits [9,10,11].

In wt animals, hippocampal dentate granule cells are densely packed into a C-shaped cell layer and display their characteristic cone-shaped dendritic arbor that is oriented perpendicular to the hippocampal fissure. In *reeler*, dentate granule cells are not arranged in a compact cell layer but are rather dispersed, including ectopic granule cells in the dentate hilar region. Moreover, in *reeler,* dentate granule cell dendrites are poorly developed and are often misoriented [3,12].

The effect of reelin deficiency on neuronal dendritic growth has been intensely investigated. Thus, pyramidal cells in the *reeler* neocortex were analyzed already at early embryonic stages [13]. The supplementation of recombinant reelin to primary cultures of cortical neurons was found to accelerate dendritic growth by acting on microtubules and to increase dendritic branching [14,15]. Cultured embryonic *reeler* hippocampal neurons showed reduced dendritic growth and complexity when compared to neurons derived from wt animals. Conversely, addition of recombinant reelin to *reeler* hippocampal neurons prepared from E18 brains induced an increase in dendritic growth and dendritic branching [16].

To unravel the mechanisms that underlie reelin function, several cell type specific or conditional reelin knock out (Reln^cKO^) mice were generated. These mice have been used as model systems to study reelin functions, apart from *reeler*. For instance, knock out of reelin specifically in CR cells resulted in an almost normal morphology of the hippocampus proper, with some differences in the CA1 region, such as ectopic pyramidal cells in the stratum radiatum, reminiscent of the ectopic CA1-pyramidal cells seen in the *reeler* hippocampus [3,17]. In turn, in the CR cell-specific reelin knock out, dentate granule cells were less compactly delineated than in wt, with a shorter or missing infrapyramidal blade [18], while Reln^cKO^ in GABAergic interneurons, which express reelin at postnatal stages, resulted in a wt-like hippocampal morphology, and the mice showed normal behavior [18,19]. Overexpression of reelin induced faster neuronal migration, as well as increased granule cell dendritic length and branching [20,21]. In turn, induction of reelin deficiency in Reln^cKO^ mice at the age of two months did not change the hippocampal morphology or the behavior of these animals [22].

In contrast to Reln^cKO^ in mice aged two months [22], Reln^cKO^ in cortical organotypic slice cultures (OTCs) prepared at postnatal day 0 (P0) led to dendritic hypertrophy of proximal dendritic segments of parvalbumin positive (PV+) interneurons and to distal dendritic tree hypertrophy of both neuropeptide Y and calretinin-positive interneurons after 10 days in vitro (DIV), in line with the earlier observation of cortical interneuron hypertrophy in *reeler* [23,24]. Similarly, in Reln^cKO^ OTCs from the medial entorhinal cortex (MEC), reelin-expressing layer II stellate neurons (ocean cells) displayed dendritic hypertrophy, and MEC layer II pyramidal cells (island cells) not expressing reelin developed hypertrophic proximal apical dendritic segments at 10 DIV [25]. In the present study, we asked whether Reln^cKO^ in vivo during the early postnatal period might similarly cause malformations of hippocampal neurons.

## 2. Methods

### 2.1. Animals

The generation of the conditional knock out reelin mouse line (Reln^cKO^) has been described earlier [22]. Shortly, Reln^flox/flox^ mice were crossed with hemizygous tamoxifen-inducible Cre recombinase-expressing mice (CAG-Cre^ERT2^) [26]. For the experiments, only Reln^flox/flox^ CAG-Cre^ERT2^ male mice were selected and then crossed with Reln^flox/flox^ female mice to generate Reln^flox/flox^ wild type (wt) and Reln^flox/flox^ CAG-Cre^ERT2^ (Reln^cKO^) siblings, as verified by PCR. The cKO mouse line ubiquitously expresses a fusion protein comprising Cre recombinase and a mutated form of the estrogen receptor (Cre-ERT2). The administration of tamoxifen leads to the activation of Cre and the knockout of the floxed Reelin gene [27].

Animals were housed in a standard 12 h light cycle and fed ad libitum with standard mouse chow. Newborn pups were administered 20 mg/mL tamoxifen (Sigma Aldrich, Darmstadt, Germany) in a solution of 10% ethanol and 90% corn oil (Sigma Aldrich, Darmstadt, Germany) via oral gavage from postnatal day 1 (P1) for a period of five consecutive days.

In an earlier study, we have already shown that in both in vivo experiments and in vitro slice culture experiments, the administration of tamoxifen results in an elimination of reelin immunostaining in the somatosensory cortex, whereas four weeks after tamoxifen induction, Reln^cKO^ resulted in a quantifiable amount of reelin-positive cells in the hippocampus, and a very small amount of reelin protein was still detectable by Western blot. This suggests that residual undegraded reelin remains in the extracellular matrix [27], but it also points to an incomplete Reln^cKO^, with some remaining reelin expressing cells that are unaffected by tamoxifen administration.

### 2.2. Genotyping of the Animals

Genotyping, based on our previously described protocol [23,25], was slightly modified as follows: Briefly, DNA was isolated from the tail or ear of the animals. The tissue was placed into 20.5 µL of dilution buffer including DNA Release Additive (from Phire Animal Tissue Direct PCR Kit; Thermo Fisher Scientific; Waltham, MA, USA) and incubated for 5 min (ear) or for 15 min (tail) at 98 °C. The PCR reaction was performed in a total volume of 25 µL, including 2 µL of DNA sample, Taq Polymerase, GoTaq Buffer, primers, and dNTPs. For genotyping, we used the following primers: wild type mice, forward primer 5′-ATAAACTGGTGCTTATGTGACAGG-3′, reverse primer 5′-AGACAATGCTAACAACAGCAAGC-3′ (450 bp). Reln^flox/flox^ mice, forward primer 5′-GCTCTGGCCAAGCTTTATC-3′, reverse primer 5′-CGCGATCGATAACTTCGTATAGCATAC-3′ (1200 bp). For detection of CAG-Cre^ERT2^, forward primer 5′-ATTGCTGTCACTTGGTCGTGG-3′, reverse primer 5′-GGAAAATGCTTCTGTCCGTTTGC-3′ (200 bp). The samples were run on a 1.5% agarose gel including 0.004% Midori Green Advance DNA Stain (NIPPON Genetics EUROPE; Düren, Germany). Gels with fluorescent DNA bands were evaluated under UV light.

### 2.3. Western Blot from Brain Tissue Two and Four Weeks Postnatally

Animals were sacrificed by decapitation after anesthesia with Isofluran CP^®^ (CP Pharma, Burgdorf, Germany), then the skull was opened, and the whole brain was removed and placed into chilled Gey’s Balanced Salt Solution (GBSS) containing 25 mM D-glucose. The cortex, hippocampus, and cerebellum were frozen in liquid nitrogen and stored at −80 °C until use. Hippocampal tissue was then homogenized in ice-chilled urea lysis buffer (6 M urea and 12 mM magnesium-acetate in 100 mM Tris-HCl), then centrifuged at 10,000× *g* for 15 min to remove debris and nuclei. The supernatant was then diluted with Laemmli buffer before separation of proteins according to size on a 7.5% or 10% SDS-PAGE gel. The proteins were transferred onto nitrocellulose membrane (pore size: 0.2 µm), then incubated with the following antibodies using the PURITY^TM^ detection system (Vilber Lourmat GmbH, Eberhardzell, Germany) overnight at 4 °C: anti-reelin G10 (Merck Millipore, Darmstadt, Germany, MAB5364) 1:500, anti-Cre (Millipore, 69050-3) 1:1000, and anti-β-tubulin (Sigma-Aldrich, T4026). The next day, proteins were visualized using the PURECL^TM^ substrates (Vilber Lourmat GmbH, Eberhardzell, Germany) using a Vilber Fusion FX imaging system.

### 2.4. Immunocytochemistry of Whole Brain Sections

Whole brains of wt or Reln^cKO^ mice were dissected as described above and fixed directly after removal in 4% paraformaldehyde for 72 h at 4 °C on a shaker. Subsequently, whole brains were washed for 24 h in PBS, then embedded in 2% agarose and mounted with instant adhesive on the sample holder of a vibratome for cutting. The brains were cut with a vibratome (Leica VT 1000 S, Wetzlar, Germany) into 50 µm thick frontal sections and washed with 0.1 M PBS. Free-floating brain sections were incubated with 10 mM citrate buffer (pH 6) at 60 °C for 1 h for antigen retrieval, permeabilized, and blocked with 0.2% Triton X-100 in blocking solution (5% normal goat serum (NGS); 2.5% bovine serum albumine (BSA); in 0.1 M PBS) for 1 h at RT. Next, brain sections were incubated in blocking solution containing 0.1% Triton X-100 with anti-Reelin CR50 antibody (D223-3, MBL; 1:1000) free floating on a shaker at 4 °C overnight. After three washing steps, each 10 min with 0.1 M PBS, the fluorescent secondary antibody (ThermoFisher, goat-anti-mouse Alexa-Fluor 568) was added and incubated for 1.5 h at RT. Finally, brain sections were counterstained with the nuclear dye 4′,6-Diamidin-2-phenylindol (DAPI) and mounted on glass slides with ROTI^®^Mount FluorCare (Carl Roth, Karlsruhe, Germany). Pictures were captured with a confocal spinning disc microscope (Visitron Systems GmbH, Puchheim, Germany). Reelin immunoreactive cells were counted manually on blinded genotype sections. Cell numbers were normalized to the total number of cells (DAPI-positive nuclei) within a picture frame.

### 2.5. Golgi Staining and Neurolucida Reconstructions

Animals aged 1 month were anesthetized using Isofluran CP^®^ (CP Pharma) then quickly decapitated. Brain samples were carefully removed from the skull, dipped into GBSS containing 25 mM D-glucose, and the two brain hemispheres were separated. Golgi-Cox impregnation was performed using FD Rapid GolgiStain^TM^ Kit (FD NeuroTechnologies Inc., Columbia, MD, USA) according to the company’s instructions. Shortly, hemispheres were placed into solution A/B for 2 weeks, then into solution C for 72–80 h. Sagittal Vibratome sections measuring 90 µm in thickness were cut, dried for 2 days, then stained with solution D/E. Following dehydration, sections were mounted with Eukitt^®^ Quick-hardening mounting medium (Sigma Aldrich). Neuron reconstructions were performed with Neurolucida software (version 11.08.2; MBF Bioscience, Willistin, VT, USA) using an Olympus Axiovert 200M microscope (Olympus, Tokyo, Japan) equipped with a QImaging Retiga 1350 B Camera (Teledyne QImaging, Surrey, BC, Canada) with a 63× objective according to the following selection criteria: The cell body and the proximal dendritic segments are well impregnated, and the neurons are clearly visible with minimal background. A minimum of 30 neurons per cell type (interneurons, granule cells, and pyramidal cells) were reconstructed and then divided into subgroups.

Representative pictures of the Golgi-stained neurons were made with an Olympus BH2 microscope using cellSens Entry software (version 3.2; Olympus, Tokyo, Japan). Sholl analysis was performed with a starting radius of 10 µm and increased stepwise by 10 µm until a radius of 40 µm was reached [28]. The following morphological parameters were analyzed using the NeuroExplorer software (version 11.08.2) provided to the Neurolucida system (MBF Bioscience, Willistin, VT, USA): soma area, perimeter, largest and smallest diameters of the soma, number of dendritic intersections, dendritic average segment diameter, segment length, segment surface area (2D), and segment volume (3D); distance from the soma, reflecting the radius of the Sholl rings. Mean dendritic intersections in the Sholl analysis refers the total number of intersections in the sphere, where mean dendritic length, surface area, and volume refer to the total dendritic length, surface, and volume passing through the set Sholl ring. Bifurcation angle and tortuosity, reflecting dendritic meandering, are defined as the ratio of length and the length of the straight line between the beginning and the end of a branch (tortuosity = 1 denotes a straight branch while increasing values denote more tortuous branches); branch angle and tortuosity parameters were analyzed according to the tree order, up to order number 3. For analysis of dendrites, only dendritic segments that were completely reconstructed were included in the analysis. For further details on the definitions of the parameters, please refer to references [29,30].

### 2.6. Statistics

Statistical analysis was performed using GraphPad Prism 10 or Excel (Microsoft, Version 2510) software. After normality testing, the Mann–Witney U-test or Student’s *t*-test was used to determine significance. Values are presented as mean ± standard error of the mean (SEM). The significance level was set to ≤0.05.

## 3. Results

### 3.1. Assessment of Reelin Expression After Early Postnatal Reln^cKO^

To assess the loss of reelin expression after early postnatal Reln^cKO^, brain tissue from Reln^cKO^ mice was analyzed by Western blotting and immunohistochemistry using the reelin-specific antibody CR50, as previously described (see Section 2 and ref. [27]). In Western blots, a distinct band representing the 180 kDa proteolytic fragment of reelin was clearly detectable for wt animals aged 14 or 28 days, respectively. For Reln^cKO^, either no signal or, in some blots, a very faint 180 kDa signal was detected, suggesting that either reelin in the extracellular matrix had not been completely proteolytically degraded after Reln^cKO^, or that some of the cells still expressed reelin. However, in all Reln^cKO^ samples analyzed by Western blot, a distinct Cre-ERT2 signal at approximately 70 kDa was detectable, reflecting the functionality of tamoxifen-induced Cre-activation (Figure 1D). Next, we analyzed the number of reelin-expressing hippocampal neurons by immunohistochemistry in brain sections from Reln^cKO^ mice aged 4 weeks. While we found numerous reelin-expressing cells in wt control tissue sections, only few reelin-positive cells were detected in Reln^cKO^ tissue (Figure 1A,B). We conclude that in these few scattered Reln^cKO^ hippocampal neurons, tamoxifen-induced recombination was either not successful, or, alternatively, already transcribed and translated reelin might have been stored over several days in these cells but was not secreted. We conclude that Reln^cKO^ induced by tamoxifen feeding was successful, but likely not with 100% efficiency, and consequently, small amounts of reelin and a markedly reduced number of reelin-expressing cells were still detectable in the hippocampal tissue of Reln^cKO^ mice aged 4 weeks.

### 3.2. Multipolar Interneurons Develop Dendritic Hypertrophy After Early Postnatal Reln^cKO^

A total number of 31 (control) and 32 (Reln^cKO^) interneurons were reconstructed in wt and in the Reln^cKO^ animals, respectively. Inhibitory GABAergic neurons can be divided into a variety of subtypes according to their morphological and electrophysiological properties, and specific protein expression markers [31,32]. We chose the simplest morphological feature to categorize these neurons: bitufted neurons with dendrites mainly originating from the two poles of the cell body (*n* = 12–17) and multipolar neurons (*n* = 11–17) [33]. We excluded mossy cells (*n* = 3) and interneurons, where the dendritic tree is concentrated close to the soma (*n* = 3).

Multipolar interneurons did not show differences in the perimeter or the soma area (Figure 2E,F) in the tortuosity of the first and second dendritic branches (Figure 2G,H). Sholl analysis revealed an increased number of intersections and increased dendritic length, reaching significance at a 30 µm dendritic distance for the soma (65.04 ± 7.00 µm in wt vs. 97.79 ± 11.00 µm Reln^cKO^, *p* = 0.0405) (Figure 2J). These results confirm proximal dendritic hypertrophy of these reelin-deficient interneurons, as described earlier in the neocortex and entorhinal cortex [23,25].

When analyzing bitufted interneurons, no difference in the soma parameters could be detected (Figure 3E,F). In turn, an increase, though not reaching significance, in the tortuosity of the first and second dendritic segments could be observed (1.06 ± 0.02 in wt vs. 1.09 ± 0.02 in Reln^cKO^, *p* = 0.0512 and 1.16 ± 0.03 in wt vs. 1.25 ± 0.04 in Reln^cKO^, *p* = 0.0667, respectively) (Figure 3G,H). Sholl analysis revealed no differences in the dendritic segments of Reln^cKO^ animals when compared to wt (Figure 3I–L).

In sum, multipolar interneurons appear to be more affected by Reln^cKO^ than bitufted interneurons.

### 3.3. Dentate Granule Cells Develop Atrophic Proximal Dendritic Segments and Smaller Cell Bodies After Early Postnatal Reln^cKO^

A total of 31 (wt) and 32 (Reln^cKO^) dentate granule cells were reconstructed in control and Reln^cKO^ animals, respectively. Representative Golgi staining micrographs were captured (Figure 4A,C) and paired with the reconstructed neurons (Figure 4B,D). First, cell body analysis was performed. Our measurements revealed significantly reduced granule cell soma perimeters (46.95 ± 0,87 µm in wt vs. 44.21 ± 0,86 µm in Reln^cKO^, *p* = 0.0064) and areas (152.88 ± 5.94 µm^2^ in wt vs. 132.63 ± 4.39 µm^2^ in Reln^cKO^, *p* = 0.0055) in Reln^cKO^ animals when compared to wt mice (Figure 4E,F). When investigating the minimal and maximal cell body diameter, a significant reduction in the maximal diameter (17.53 ± 0.38 µm in wt vs. 16.36 ± 0.38 µm in Reln^cKO^, *p* = 0.0152) was detected in the absence of reelin, indicating formation of an abnormal shape of the granular cell soma (Figure 4G). When analyzing the dendritic segments, we found increased tortuosity of the first dendritic branch/es in the absence of reelin (1.07 ± 0.01 in wt vs. 1.14 ± 0.03 in Reln^cKO^, *p* = 0.0231, Figure 4H), leading to the conclusion that these dendritic segments tend to be not as straight in Reln^cKO^ as in the wt animals. Sholl analysis of the dendrites revealed a significantly decreased number of dendritic intersections after Reln^cKO^, beginning at a distance of 10 µm (Figure 4I). Moreover, decreased dendritic length and dendritic surface area were observed, beginning at a distance of 20 µm in Reln^cKO^ mice (Figure 4J,K), whereas the mean average dendritic diameter was increased in the mutant animals (Figure 4L), reflecting dendritic atrophy of dentate granule cells. In summary, the early postnatally induced reelin deficiency severely affects both the cell body and dendrites of dentate granule cells, resulting in a perikaryon with a smaller circumference and a more elongated shape when compared to the perikarya of wt animals, and in significant changes in the morphology of proximal dendrites, including twisted proximal dendritic segment and dendritic atrophy.

Taken together, we found that granule cells in Reln^cKO^ develop a smaller soma with twisted first dendritic segments and atrophy of the proximal dendritic tree.

### 3.4. Dendritic Malformations of Hippocampal Pyramidal Cells After Reln^cKO^ Depend on the CA-Region

We reconstructed a total of 33 and 30 pyramidal neurons in the wt and Reln^cKO^ groups, respectively. After the reconstructions, we sorted them according to their regional position in the pyramidal cell layer: CA3 (*n* = 9–10), CA2 (*n* = 9), and CA1 (*n* = 8–9). A total of 6 and 3 pyramidal cells were excluded from the analysis in the wt and Reln^cKO^ groups, respectively, through their localization in the presubiculum, subiculum, or entorhinal cortex. Apical and basal dendritic segments were investigated separately. While the soma parameters of the CA3 pyramidal cells from Reln^cKO^ animals did not show any difference when compared to control (Figure 5E,F), the apical dendritic angle of these neurons at dendritic tree order 3 increased in Reln^cKO^ mice (35.6 ± 4.57 in wt vs. 48.65 ± 1.14 in Reln^cKO^, *p* = 0.0077) (Figure 5H), reflecting a change in the orientation of these segments. The basal dendritic tree of Reln^cKO^ animals remained normal as in wt, which is in contrast to basal dendrites of pyramidal island cells in the entorhinal cortex of Reln^cKO^ [25]. Sholl analysis revealed no difference in the mean dendritic intersections or in the mean dendritic length of the apical and basal dendritic segments. Only the mean apical dendritic average diameter decreased at 30 µm dendritic length in the Reln^cKO^ animals (4.26 ± 0.44 µm in wt vs. 2.81 ± 0.44 µm in Reln^cKO^, *p* = 0.0325) (Figure 5L), whereas it remained unchanged in the basal dendritic branches. The dendritic volume was decreased at the basal dendritic segments at 30 µm distance (106.48 ±17.19 µm^3^ in wt vs. 67.4 ± 16.10 µm^3^ in Reln^cKO^, *p* = 0.0435) (Figure 5O). Although CA3 pyramidal cell bodies were normal in the Reln^cKO^ group, the orientation of their apical dendrites changed in the absence of reelin, with subtle (statistically non-significant) atrophy in the apical and basal dendritic segments.

We found smaller cell body areas in the case of CA2 pyramidal cells in Reln^cKO^ animals (262.85 ± 14.16 µm^2^ in wt vs. 213.72 ± 14.30 µm^2^ in Reln^cKO^, *p* = 0.0266), where the perimeter of the cell bodies remained unchanged (Figure 6E,F). Concerning CA2 pyramidal cells, a decreased apical dendritic angle at the 2nd order segments could be found in Reln^cKO^ mice (51.27 ± 3.66 in wt vs. 35.38 ± 6.16 in Reln^cKO^, *p* = 0.0376) (Figure 6G), while the basal dendritic segments remained unchanged. Sholl analysis of the apical dendritic segments revealed atrophy, gaining significance at 30 µm soma distance in dendritic surface area (159.49 ± 18.03 µm^2^ in wt vs. 91.27 ± 13.40 µm^2^ in Reln^cKO^, respectively, *p* = 0.0078) (Figure 6K), whereas significant changes in dendritic volume at 30 µm (149.33 ± 23.58 µm^3^ in wt vs. 74.64 ± 14.43 µm^3^ in Reln^cKO^, *p* = 0.0157) and at 40 µm (104.88 ± 19.25 µm^3^ in wt vs. 56.93 ± 9.78 µm^3^ in Reln^cKO^, *p* = 0.0411) (Figure 6L), and dendritic length (19.36 ± 4.41 µm in wt vs. 9.23 ± 1.16 µm in Reln^cKO^, *p* = 0.0030 at 30 µm and 18.13 ± 2.45 µm in wt vs. 15.32 ± 5.95 µm in Reln^cKO^, *p* = 0.0300 at 40 µm) (Figure 6J) were detected at a distance of between 30 µm and 40 µm from the soma. Investigation of the basal dendritic segments revealed the same tendency, but only the average dendritic diameter at 20 µm soma distance decreased significantly (1.68 ± 0.16 in wt vs. 1.22 ± 0.13 in Reln^cKO^, *p* = 0.0403) (Figure 6R).

No differences could be found in any of our investigated parameters of CA1 pyramidal cells. Accordingly, the morphology of CA2 pyramidal cells is more affected by the early postnatal Reln^cKO^ when compared to pyramidal cells of the CA1- and CA3-region.

To summarize, we found that all subtypes of hippocampal neurons display morphological malformations after early postnatal Reln^cKO^; however, the grade of the deformations varies between the cell types. Thus, dentate granule cells are most severely affected, malformations of pyramidal cells are less severe and depend on their position in the subregions CA1, CA2 or CA3, while interneuron morphology is only mildly affected.

## 4. Discussion

In the present study, we investigated the effect of conditionally induced reelin deficiency on hippocampal neurons when induced immediately after birth in an in vivo mouse model. It was reported earlier that after Reln^cKO^ at the age of two months, no morphological defects were detected in the hippocampus. In turn, we had already shown that early postnatal reelin deficiency leads to dendritic hypertrophy of some interneuron subtypes in the neocortex, and of both layer II stellate and pyramidal cells in the entorhinal cortex [23,25,27]. Moreover, dendritic hypertrophy of interneurons has been reported in the neocortex of *reeler* [3,16]. For the present study, we induced postnatal reelin deficiency early after birth and analyzed the morphological parameters of hippocampal neurons in these mice when aged four weeks.

### 4.1. Only Mild Malformations of Interneurons After Early Postnatal Reln^cKO^

Maturation of interneurons in the cerebral cortex undergoes a long postnatal period, lasting up to four weeks [34,35]. Correct wiring of these neurons is important for the excitatory–inhibitory balance. For instance, the basket cells and chandelier cells with their synapses on the soma, proximal dendrites, and axon initial segments of different neuron types play a crucial role in controlling their excitability [35,36]. These neuron types are also positive for the calcium-binding protein parvalbumin [37]. Other neurons, like bipolar and bitufted cells, target mainly the basal part of pyramidal cells [33]. In the cortex, neuropeptide Y-positive bitufted interneurons, together with horizontal cells and Martinotti cells, are known to express reelin [38].

Our results concerning the morphology of multipolar neurons, most of which express parvalbumin, are in line with earlier studies, where dendritic hypertrophy of parvalbumin-positive interneurons in the somatosensory cortex was observed [23]. However, we did not observe significant changes in the proximal dendritic segments of bitufted interneurons [23,33].

### 4.2. Of All Hippocampal Neuronal Cell Types, Dentate Granule Cells Display the Most Severe Malformations After Early Postnatal Reln^cKO^

Correct radial migration of dentate granule cells depends on reelin, secreted by hippocampal CR cells, which are located in the outer molecular layer of the dentate gyrus. The results of our experiments show that early postnatally induced reelin deficiency substantially impairs the morphological differentiation in particular of dentate granule cells, resulting in a smaller soma perimeter, soma area, and an atrophic proximal dendritic tree with increased average diameter, whereas the first order of the dendritic processes developed a tortuous shape. Morphological changes in the dentate granule cells could be explained by the reelin-dependent maturation of this cell type, but alternatively, different inhibition onto the soma and the proximal dendrites coming from the interneurons could be causally linked to the smaller size of dentate granule cells.

The extent of cell type-specific morphological defects that we observed in our experiments after early postnatal Reln^cKO^, i.e., granule cells were more severely affected than pyramidal cells and interneurons, might be related to the different, developmentally regulated birth dates and differentiation periods of these neurons. While pyramidal cells of the Ammon’s horn, similarly to GABAergic non-pyramidal interneurons, originate prenatally from around E10 to E18, the majority of dentate granule cells is generated postnatally. Newborn granule cells migrate and differentiate under control of reelin to form the dentate gyrus [39,40,41,42,43]. This late appearance of granule cells may explain why the morphology of dentate granule cells is more affected than that of hippocampal pyramidal cells after postnatally induced Reln^cKO^. In turn, we previously described neuronal malformations in the entorhinal and neocortex of Reln^cKO^ mice [23,25], i.e., cells that migrate prenatally, indicating that specific neuronal cell types in different cortical areas are more vulnerable to loss of reelin than others. We did not observe granule cell dispersion in the hilar region, which is however characteristic of the dentate gyrus of *reeler*. What could be the reason for the absence of granule cell dispersion? In previous and recent experiments, we have already shown that tamoxifen-induced Reln^cKO^ is not immediately followed by complete disappearance of the reelin that had already been secreted into the extracellular matrix (ECM) [25,27]. This remaining reelin, which is eventually degraded, appears to be sufficient to orchestrate the formation of a compact granule cell layer but not sufficient to promote proper dendritic differentiation. In this context, it should be noted that heterozygous *reeler* do not show neuronal migration defects in spite of 50% reduced reelin expression, supporting the interpretation that reduced amounts reelin in the ECM of the developing hippocampus are still sufficient to control correct neuronal positioning in the dentate gyrus [44,45]. Of note, reelin is also known to modulate adult hippocampal neurogenesis [20,46,47]. A more detailed analysis of newborn dentate granule cells at early and adult stages will be required in the future to assess to what extent effects on neurogenesis might contribute to neuronal malformations after postnatal Reln^cKO^.

### 4.3. Malformations of Pyramidal Cells After Early Postnatal Reln^cKO^ Differ Between Subregions CA1, CA2, and CA3

Concerning the pyramidal cells, we found that after early postnatal conditionally induced reelin deficiency CA1 pyramidal cells displayed a normal, wt-like morphology, while in the same tissue samples, CA2 and CA3 pyramidal cells showed signs of dendritic atrophy, mainly in the apical dendritic segments, reflected by an altered dendritic angle at the 2nd or 3rd tree order. Moreover, CA2 pyramidal neurons had a smaller cell body, reminiscent of the reduced size of granule cells after conditionally induced reelin deficiency. The observed differences in dendritic atrophy regarding the different pyramidal cell types, CA1-, CA2-, or CA3-pyramidal cells, respectively, suggest that reelin effects are pyramidal cell subtype specific. Since the birth dates of CA2- and CA3-pyramidal cells are similar to those of CA1 pyramidal cells [48], differential reelin effects related to the pyramidal cell birth dates seem unlikely.

### 4.4. Stage- and Cell Type-Specific Reelin Knock Out Strategies Reveal Different Reelin Functions

Previous studies had shown that Reln^cKO^, when restricted to interneurons, only mildly affected hippocampal development, much in contrast to *reeler*, which lacks reelin expression in both CR-cells and interneurons [18,19]. In turn, inactivation of reelin expression, when restricted specifically to CR cells, caused dramatic morphological changes in the dentate gyrus, while defects in the hippocampus proper were only mild [18]. Moreover, cell type-specific Reln^cKO^ in either reelin-secreting CR-cells or GABAergic interneurons suggests that these cell types cooperate in secreting reelin to prevent invasion of cortical neurons into the marginal zone, the future layer I [18]. Early postnatal Reln^cKO^ induced hypertrophy of hippocampal (present study) and neocortical GABAergic interneurons [23] but no apparent neuronal migration defects. Overexpression of reelin, in turn, was shown to increase adult neurogenesis, the number of synaptic contacts, and to induce hypertrophy of dendritic spines, paralleled by increased long-term potentiation responses [20]. Conditionally induced Reln^cKO^ in all reelin-expressing cells at the age of two months did not affect brain lamination [22,23].

### 4.5. Reduced Reelin Expression and Neuropsychiatric Disorders

Decreased reelin expression levels in the neocortex, entorhinal cortex, and hippocampus, combined with altered neurotransmitter receptor functions or impaired reelin signaling, were observed in humans suffering from a variety of different neurological disorders, including autism, schizophrenia, and Alzheimer’s disease [49,50,51,52,53,54]. In line with these observations, Golgi-Cox staining revealed dendritic atrophy of dentate granule cells in post-mortem brains of patients suffering from Alzheimer’s disease [55]. Vice versa, a gain-of-function mutation in the human reelin gene (RELN-COLBOS) that provides resilience to autosomal dominant Alzheimer’s disease has recently been described [56].

## 5. Conclusions

In summary, our present findings underline that morphological changes induced by early postnatal loss of reelin are dramatic when compared to late (at least 2 months after birth) induced reelin deficiency, which does not appear to entail morphological changes, while electrophysiological parameters, such as LTP, appear to be affected [22]. The results of the present study regarding hypertrophy of hippocampal multipolar interneurons after tamoxifen-induced Reln^cKO^ are in line with our previous findings, i.e., hypertrophy of multipolar interneurons in the somatosensory and entorhinal cortex [23,25]. Moreover, we found that granule cells had a reduced soma size and developed an atrophic dendritic tree in Reln^cKO^. Pyramidal cells of the Ammon’s horn showed apical dendritic atrophy in the CA3 and CA2 regions, with aberrant orientation of the dendritic trees. CA1 pyramidal cells, in turn, were less prone to malformation, as we did not detect changes in their morphology in Reln^cKO^. We consider the differential malformations of hippocampal, and, as previously reported, of entorhinal and neocortical neurons after early postnatally induced Reln^cKO^ as an important finding, since these malformations may reflect a transient morphological plasticity which, on the one side, endangers the differentiating neurons in the cerebral cortex to suffer from morphological malformations but, on the other side, may allow neurons during a limited time window to adapt their dendritic morphology to specific neuronal network requirements.

### Methodological Considerations

In the present study, we focused on the question whether Reln^cKO^, when induced at early postnatal stages, causes morphological changes in hippocampal neurons. Our methodological approach, i.e., reconstruction of Golgi-stained neurons in tissue sections and their quantitative analysis, has, like any other experimental approach, its inherent methodological limitations. Thus, in particular, long distal Golgi-stained dendritic segments might be transected during the tissue sectioning procedure, thereby disclosing precise quantification of the most distal dendritic length and/or branching patterns. However, we want to emphasize that the main goal of our experiments was a proof-of-principle study, i.e., to demonstrate that postnatally induced Reln^cKO^ causes malformations of hippocampal neurons when induced during a critical period early after birth, but not when induced later, at the age of two months, as reported earlier. To draw this conclusion, a complete description of all induced neuronal malformations after Reln^cKO^ is not required.

## Figures and Tables

**Figure 1 biomolecules-15-01662-f001:**
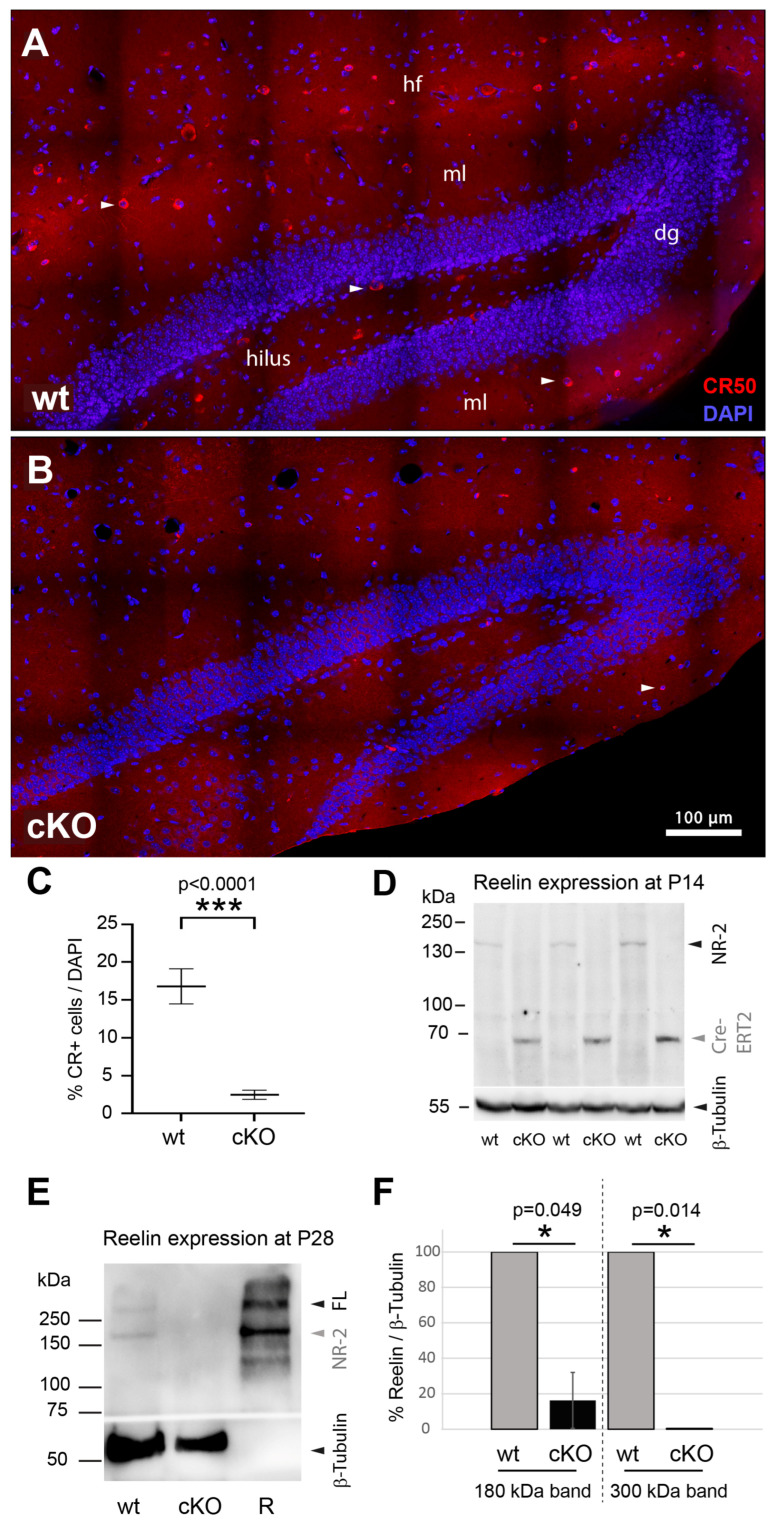
Loss of reelin expression after early postnatal tamoxifen-induced Reln^cKO^. Representative four-week-old, stitched reelin immunostained hippocampal brain sections captured with a 60× objective ((**A**,**B**) wt mice *n* = 6, Reln^cKO^ mice, *n* = 6). In wt, numerous red, reelin-positive cells (anti-reelin (CR50) antibody staining; white arrowheads exemplary mark reelin positive cells) are detectable in the molecular layer (mL), hilus of the dentate gyrus (dg), and along the hippocampal fissure (hf) (**A**). Cell nuclei were counterstained with the nuclear dye DAPI (blue). Reelin-positive immunostaining (red) is largely lost after tamoxifen-induced Reln^cKO^ (**B**). However, none of the animals exhibited a complete loss of reelin immunostaining, and a small number of reelin positive cells was still detectable in the brain sections of Reln^cKO^ mice ((**B**) white arrowhead). Quantification of reelin expression by counting reelin immunopositive cells normalized to the total number of DAPI-positive cells within individual brain sections of wt vs. Reln^cKO^ confirmed a significant reduction, but not a complete loss of reelin-positive cells (*p* < 0.0001) (**C**). Western blot analysis of brain tissue from two-week-old wt animals (P14, *n* = 3) showed a clear 180 kDa reelin band (NR-2 fragment), whereas after tamoxifen-induced Reln^cKO^ (cKO) the 180 kDa reelin band is no longer detectable. However, a distinct Cre-ERT2 band of ~70 kDa is visible, indicating a functional Cre-loxP mediated recombination in the Reln^cKO^ (**D**). Four weeks (P28) after Reln^cKO^, the signal for the NR-2 reelin fragment (~180 kDa) as well as for the full-length reelin (FL) (~300 kDa) are markedly reduced. The correct size of the reelin signals is verified by recombinant reelin (R), serving as a positive control (**E**). Densitometric quantification of Western blot reelin signals at P28 confirms a significant reduction in the reelin-positive 180 kDa and 300 kDa signals in Reln^cKO^ compared to wt, reflecting successful Cre-loxP-mediated Reln^cKO^ (**F**). * *p* ≤ 0.05. *** *p* < 0.001.

**Figure 2 biomolecules-15-01662-f002:**
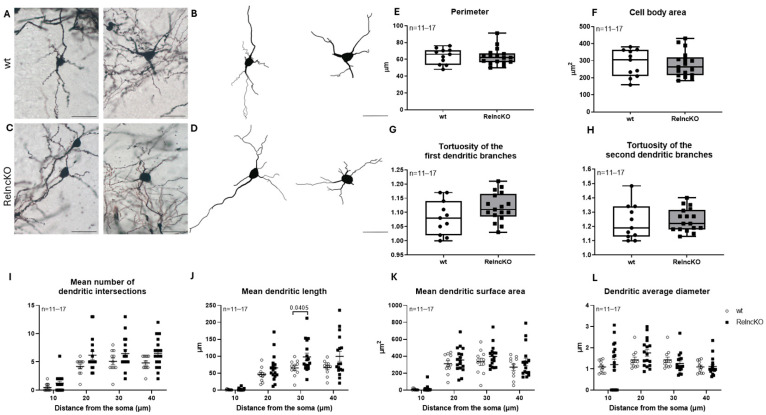
Dendritic alterations in multipolar interneurons after early postnatal Reln^cKO^. Representative micrographs of multipolar interneurons (**A**,**C**) and Neurolucida reconstructions (**B**,**D**) in wt and Reln^cKO^ hippocampus. No change in cell body perimeter (**E**) or area (**F**) could be observed in the absence of reelin. Also, no difference could be found in the tortuosity of the first and second dendritic branches of these neurons (**G**,**H**). Sholl analysis revealed increased tendency of dendritic intersections (**I**) and increased dendritic length, reaching significance at a soma distance of 30 µm (**J**). No significant difference could be observed in the dendritic surface area and in the dendritic average diameter (**K**,**L**). Scale bars: 50 µm.

**Figure 3 biomolecules-15-01662-f003:**
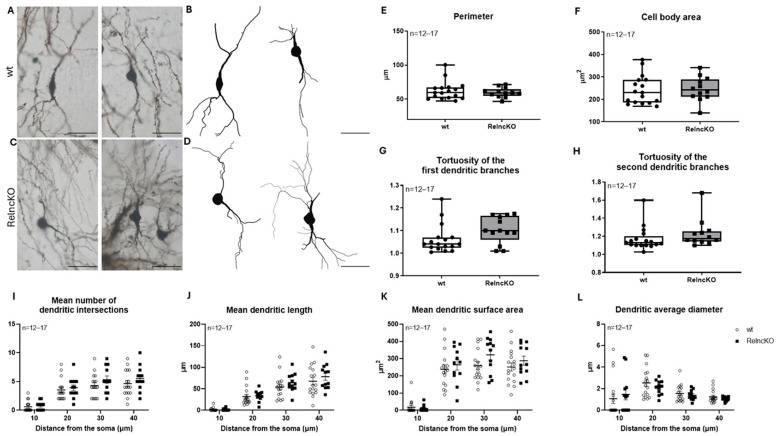
Morphology of bitufted interneurons is not significantly altered after early postnatal Reln^cKO^. Representative micrographs of bitufted interneurons (**A**,**C**) and Neurolucida reconstructions (**B**,**D**) in wt and Reln^cKO^ hippocampus. No change in cell body perimeter (**E**) or area (**F**) could be observed in the absence of reelin, but a tendency for increased tortuosity of the first and second dendritic branches could be found, however not reaching significance (**G**,**H**). Sholl analysis did not reveal changes in the dendritic segments (**I**–**L**). Scale bars: 50 µm.

**Figure 4 biomolecules-15-01662-f004:**
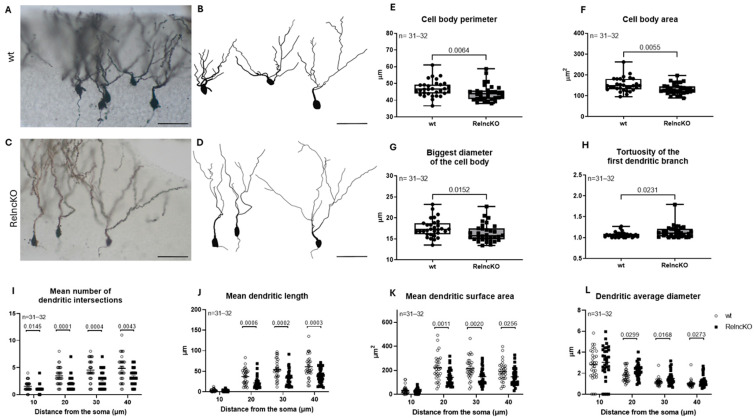
Dentate granule cells display marked somatic and dendritic malformations after early postnatal Reln^cKO^. Representative micrographs of Golgi-stained dentate granule cells combined with Neurolucida reconstructions of wild type (wt) (**A**,**B**) and conditional reelin knock out (Reln^cKO^) animals (**C**,**D**). Golgi staining of brain tissue was performed for animals aged 4 weeks. In Reln^cKO^, cell body perimeter (**E**), cell body area (**F**), and largest cell body diameter (**G**) were decreased when compared to wt. In turn, we found increased tortuosity of the first dendritic branch in Reln^cKO^ (**H**). Sholl analysis revealed fewer dendritic intersections in Reln^cKO^ compared to wt (**I**). The mean dendritic length (**J**) and the mean dendritic surface area (**K**) were also decreased, while the average diameter of the dendrites was found to be increased (**L**). Scale bars: 50 µm.

**Figure 5 biomolecules-15-01662-f005:**
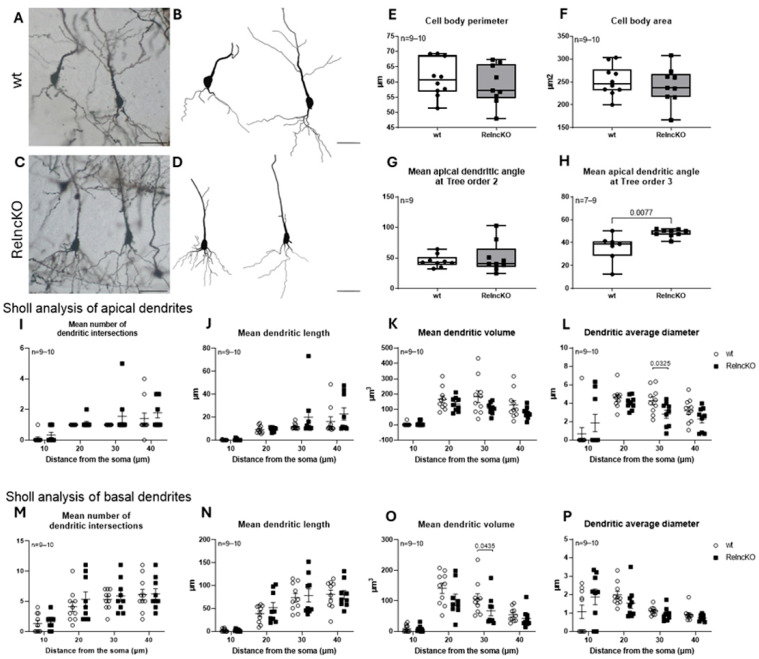
The morphology of CA3 pyramidal cell basal and apical dendrites is affected after early postnatal Reln^cKO^. Representative micrographs of Golgi-stained CA3 pyramidal cells (**A**,**C**) combined with Neurolucida reconstructions of these cells (**B**,**D**) in wt and Reln^cKO^ animals. No changes in soma perimeter (**E**) and area (**F**) were found. However, a significant increase in the apical dendritic angle at tree order 3 (**H**) could be observed in Reln^cKO^ compared to wt, but not at tree order 2 (**G**). Sholl analysis of the apical dendrites revealed significantly decreased average dendritic diameters in Reln^cKO^ at a 30 µm distance from the soma (**L**). Decreased basal dendritic volume could be observed also at 30 µm distance from the soma (**O**). No differences were found for other Sholl parameters (**I**,**J**,**K**,**M**,**N**,**P**). Scale bars: 50 µm.

**Figure 6 biomolecules-15-01662-f006:**
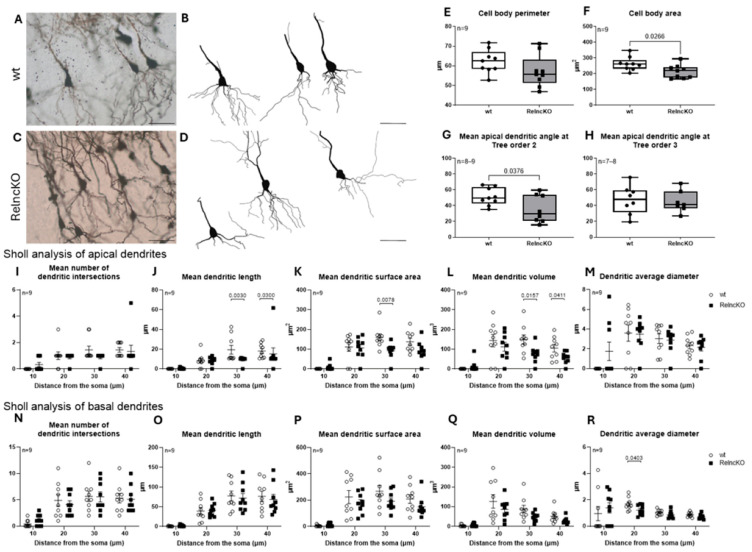
CA2 pyramidal cell morphology is altered after early postnatal Reln^cKO^. Representative micrographs of CA2 pyramidal cells (**A**,**C**) with Neurolucida reconstructions (**B**,**D**) in wt and Reln^cKO^ animals. The soma perimeter of Reln^cKO^ CA2 pyramidal cells did not differ from wt (**E**), but the soma area decreased (**F**). The apical dendritic angle decreased at the 2nd dendritic segments in the Reln^cKO^ animals compared to wt (**G**). Sholl analysis revealed a decreased mean apical dendritic length (**J**) and surface area (**K**) at 30 µm soma distance, while the mean apical dendritic volume was significantly lower at 30 µm and 40 µm soma distance (**L**). The basal dendritic average diameter was significantly decreased at 20 µm soma distance (**R**). No changes were found for other Sholl parameters (**H**,**I**,**M**–**Q**). Scale bars: 50 µm.

## Data Availability

The original contributions presented in this study are included in the article/Appendix A. Further inquiries can be directed to the corresponding author.

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
