# Peer review of "Early Postnatally Induced Conditional Reelin Deficiency Causes Malformations of Hippocampal Neurons"

_biomolecules, 2025, doi:10.3390/biom15121662_

Round 1
Reviewer 1 Report
Comments and Suggestions for Authors
The manuscript by Schneider-Lodi et al. reports findings on morphological changes in hippocampal neurons in a mouse model of conditionally induced reelin deficiency immediately after birth. The authors administered tamoxifen to newborn pups of conditional reelin KO mice (RelncKO) and subsequently analyzed Golgi-stained neuron reconstructions in wild-type versus RelncKO mice using Neurolucida software. The principal findings are (1) dentate granule cells showed the most pronounced morphological deficits in RelncKO mice in contrast to mildly affected interneurons and (2) subregion-dependent atrophy of pyramidal cells was observed in RelncKO mice.
Overall, the manuscript extends previous work and is based on a detailed quantitative analysis of hippocampal neurons using Neurolucida. The demonstration that reelin deficiency can induce marked morphological alterations in the hippocampus within 4 weeks is an important observation.
However, I have several concerns regarding the current version of the manuscript:
1. The authors performed genotyping of mice to verify DNA-level recombination. However, it is not immediately clear whether additional assays were carried out to confirm to what extent the reelin protein had been functionally lost in these experiments. This point is particularly relevant given the statement in lines 354-356 that “...tamoxifen induced RelncKO is not immediately followed by complete disappearance of the reelin that had already been secreted into the extracellular matrix.” It would strengthen the study to include experimental evidence (e.g., Western blots or IHC) that demonstrates the extent and timing of reelin depletion after tamoxifen administration.
2. The number of neurons analyzed per condition varies substantially (from as few as 9 to as many as 32). This imbalance could affect statistical power and may influence the interpretation of what constitutes a mild versus significant morphological deficit. Clarification of sampling strategy and any weighting/normalization used in statistical comparisons would be helpful.
3. The Methods section should include additional details on imaging parameters (e.g., magnification, z-step size) and the specific Neurolucida reconstruction and analysis settings employed. These parameters are important for reproducibility and for evaluating the precision of morphometric measurements.
Minor comment:
Line 20: the word ‘recently’ appears duplicated.
Author Response
REVIEWER 1
The manuscript by Schneider-Lodi et al. reports findings on morphological changes in hippocampal neurons in a mouse model of conditionally induced reelin deficiency immediately after birth. The authors administered tamoxifen to newborn pups of conditional reelin KO mice (RelncKO) and subsequently analyzed Golgi-stained neuron reconstructions in wild-type versus RelncKO mice using Neurolucida software. The principal findings are (1) dentate granule cells showed the most pronounced morphological deficits in RelncKO mice in contrast to mildly affected interneurons and (2) subregion-dependent atrophy of pyramidal cells was observed in RelncKO mice.
Overall, the manuscript extends previous work and is based on a detailed quantitative analysis of hippocampal neurons using Neurolucida. The demonstration that reelin deficiency can induce marked morphological alterations in the hippocampus within 4 weeks is an important observation.
However, I have several concerns regarding the current version of the manuscript:
- The authors performed genotyping of mice to verify DNA-level recombination. However, it is not immediately clear whether additional assays were carried out to confirm to what extent the reelin protein had been functionally lost in these experiments. This point is particularly relevant given the statement in lines 354-356 that “...tamoxifen induced RelncKOis not immediately followed by complete disappearance of the reelin that had already been secreted into the extracellular matrix.” It would strengthen the study to include experimental evidence (e.g., Western blots or IHC) that demonstrates the extent and timing of reelin depletion after tamoxifen administration.
We would like to thank the reviewer for this comment. In the revised version, we completed the description in the Methods section. We added a new Fig. (now Fig.1) showing immunohistochemistry using an antibody against reelin, control versus RelncKO and a corresponding western blot, and we added the description in the method section.
We would also like to mention that we already published RelncKO control experiments in an earlier study (see supplementary material in: Hamad et al., Reelin signaling modulates GABA(B) receptor function in the neocortex. J Neurochem 2021, 156, 589-603, doi:10.1111/jnc.14990). In that study we have shown complete loss of reelin immunostaining in RelncKO mice at P14 in the somatosensory cortex when tamoxifen was fed to newly born pups at P1 for five consecutive days (Supplementary material in Hamad et al 2021, Fig. S1A and Fig. S5 C,D). Moreover, Fig. S5 A,B in that study confirms functionality of Cre-recombinase induction by tamoxifen. In Fig. 2c of Hamad et al., we have shown that a small amount of reelin is still detectable in organotypic slice cultures by western blotting after 14 days in vitro, when reelin ko was induced with tamoxifen at the first day in vitro, slice culture preparation at P0. These earlier performed control experiments and the relevant publication (Hamad et al., 2021) are now mentioned and cited in the Methods section.
- The number of neurons analyzed per condition varies substantially (from as few as 9 to as many as 32). This imbalance could affect statistical power and may influence the interpretation of what constitutes a mild versus significant morphological deficit. Clarification of sampling strategy and any weighting/normalization used in statistical comparisons would be helpful.
We completed the Methods and Results sections and clarified the sampling strategy according to your comment. Thus, we reconstructed a minimum of 30 neurons per group, i.e. the group of interneurons, of granule cells and of pyramidal cells. In the next step, we further divided the group of interneurons into interneuron subtypes and we specified the pyramidal cells according to their morphological localization (CA1-3).
- The Methods section should include additional details on imaging parameters (e.g., magnification, z-step size) and the specific Neurolucida reconstruction and analysis settings employed. These parameters are important for reproducibility and for evaluating the precision of morphometric measurements.
We completed the Methods section according to your comments. We now included the camera type and the magnification into the Methods. The analysis was performed by using the NeuroExplorer software provided from MicroBrightField, Inc, which was purchased together with the Neurolucida software. XYZ-positions are automatically stored as absolute values in the Neurolucida System starting at a given reference point before the reconstruction (usually the middle of the cell body).
Minor comment:
Line 20: the word ‘recently’ appears duplicated.
We apologize for this mistake, we corrected the sentence.

Reviewer 2 Report
Comments and Suggestions for Authors
This manuscript titled “Early postnatally induced conditional reelin deficiency causes malformations of hippocampal neurons” presents a well-structured and comprehensive study investigating the morphological consequences of early postnatal reelin knockout in hippocampal neurons. However, several aspects could benefit from clarification, refinement, or further discussion. Below are the major concerns to improvise the manuscript.
- Redundancy in the sentence “we recently could recently show” (line 20) contains a repetition. Remove one “recently” or rephrase the sentence.
- Consider rephrasing some sentences to the active voice for better readability.
- While appropriate for a specialist audience, terms like “Neurolucida” and “Golgi-stained” could be briefly contextualized for broader accessibility.
- Some sentences are long and complex, which may hinder readability. For example: “In turn, in the CR cell-specific reelin k.o., granule cells in the dentate gyrus were less compactly delineated…” could be simplified.
- The term “reeler mutant” is repeated frequently. Consider varying phrasing or consolidating references.
- The transition from general reelin function to specific experimental models could be smoother. It would be better to use subheadings or paragraph breaks to separate the historical context, previous findings, and rationale for the current study.
- A few typo errors need to be corrected e.g., “Reeling gate” should be “reeling gait.” “Reelin k.o.” is used inconsistently; consider standardizing to “reelin knockout (KO)” throughout.
- Results are dense and could benefit from clearer subheadings or summary sentences at the end of each subsection. Consider adding a brief interpretation after each result to guide the reader.
- P-values are provided, but confidence intervals or effect sizes would strengthen the statistical interpretation. Some comparisons are described as “not significant” but still discussed in detail clarify the rationale for including these.
- Use consistent terminology for neuron types (e.g., “Reln cKO” vs. “conditional reelin knockout”). Avoid abbreviations without first defining them (e.g., “DIV” for days in vitro).
- Figures are referenced but not always described in detail. Ensure each figure is clearly explained in the text. Consider summarizing key findings in a table for quick comparison across cell types.
- In the discussion section, there is a lack of mechanistic depth, redundancy and wordiness, and speculative language. Some sentences are long and complex, which may hinder readability.
- A few references are cited multiple times for similar points (e.g., refs 3, 15, 22, 24). While this is acceptable, consider consolidating citations where appropriate.
- While the bibliography is up to date, a few more recent reviews (2024–2025) on reelin signaling or hippocampal neurogenesis could enhance the context.
Author Response
REVIEWER 2:
This manuscript titled “Early postnatally induced conditional reelin deficiency causes malformations of hippocampal neurons” presents a well-structured and comprehensive study investigating the morphological consequences of early postnatal reelin knockout in hippocampal neurons. However, several aspects could benefit from clarification, refinement, or further discussion. Below are the major concerns to improvise the manuscript.
- Redundancy in the sentence “we recently could recently show” (line 20) contains a repetition. Remove one “recently” or rephrase the sentence.
We apologize for this mistake, which was also noticed by reviewer 1, we corrected the redundancy in the sentence.
- Consider rephrasing some sentences to the active voice for better readability.
We considerably rephrased the text for better readability, see revised version.
- While appropriate for a specialist audience, terms like “Neurolucida” and “Golgi-stained” could be briefly contextualized for broader accessibility.
For a broader audience accessibility, we now replaced the term “Golgi-staining” by “silver impregnation” and introduced the better understandable term “neuron morphology reconstruction” instead of “Neurolucida”.
- Some sentences are long and complex, which may hinder readability. For example: “In turn, in the CR cell-specific reelin k.o., granule cells in the dentate gyrus were less compactly delineated…” could be simplified.
The complexity of the sentences was reduced.
- The term “reeler mutant” is repeated frequently. Consider varying phrasing or consolidating references.
We corrected this point, and now varied the phrasing.
- The transition from general reelin function to specific experimental models could be smoother. It would be better to use subheadings or paragraph breaks to separate the historical context, previous findings, and rationale for the current study.
We now introduced paragraph breaks to better separate the lines of thought.
- A few typo errors need to be corrected e.g., “Reeling gate” should be “reeling gait.” “Reelin k.o.” is used inconsistently; consider standardizing to “reelin knockout (KO)” throughout.
We corrected the typo errors and standardized the expression ”RelncKO”.
- Results are dense and could benefit from clearer subheadings or summary sentences at the end of each subsection. Consider adding a brief interpretation after each result to guide the reader.
We apologize for the dense presentation of the results. For a more clear presentation and for better understanding, each section now ends with a brief summary sentence.
- P-values are provided, but confidence intervals or effect sizes would strengthen the statistical interpretation. Some comparisons are described as “not significant” but still discussed in detail clarify the rationale for including these.
We now changed the presentation of the figures to show a detailed distribution of our data points. We no longer discuss in detail the non-significant data.
- Use consistent terminology for neuron types (e.g., “Reln cKO” vs. “conditional reelin knockout”). Avoid abbreviations without first defining them (e.g., “DIV” for days in vitro).
We now use consistent terminology. The term RelncKO is first defined on page 1 where it first appears, line 19 as “conditionally induced reelin deficiency (RelncKO)...”. From there on, the abbreviation „RelncKO “ is consistently used.
The term DIV is first defined on page 2, line 90, as “..after 10 days in vitro (DIV) ” From there on, the abbreviation „DIV“ is consistently used.
- Figures are referenced but not always described in detail. Ensure each figure is clearly explained in the text. Consider summarizing key findings in a table for quick comparison across cell types.
We now made sure that each figure is clearly explained in the text. Instead of summarizing the key findings in a table we added a short summary to each paragraph.
- In the discussion section, there is a lack of mechanistic depth, redundancy and wordiness, and speculative language. Some sentences are long and complex, which may hinder readability.
We rephrased parts of the discussion for better readability.
- A few references are cited multiple times for similar points (e.g., refs 3, 15, 22, 24). While this is acceptable, consider consolidating citations where appropriate.
We rethought the positioning of the references and adapted them where appropriate.
- While the bibliography is up to date, a few more recent reviews (2024–2025) on reelin signaling or hippocampal neurogenesis could enhance the context.
We now included two recent reviews from the years 2024 and 2025 which were not cited in the earlier version of the manuscript (Katsuyama, Y.; Hattori, M. (2024) and Valderrama-Mantilla et al. (2025)

Round 2
Reviewer 1 Report
Comments and Suggestions for Authors
The authors have fully addressed my concerns.
Reviewer 2 Report
Comments and Suggestions for Authors
NA